# Effects of Edible Insect *Tenebrio molitor* Larva Fermentation Extract as a Substitute Protein on Hepatosteatogenesis and Proteomic Changes in Obese Mice Induced by High-Fat Diet

**DOI:** 10.3390/ijms22073615

**Published:** 2021-03-31

**Authors:** Ju Ri Ham, Ra-Yeong Choi, Yongjin Lee, Mi-Kyung Lee

**Affiliations:** 1Department of Food and Nutrition, Sunchon National University, Suncheon 57922, Korea; punsu05@nate.com; 2Department of Agricultural Biology, National Institute of Agricultural Sciences, Rural Development Administration, Wanju 55365, Korea; fkdud1304@korea.kr; 3Department of Pharmacy, Sunchon National University, Suncheon 57922, Korea; yojilee@gmail.com

**Keywords:** alternative protein, hepatosteatosis, mealworm, *Tenebrio molitor* larva, obesity

## Abstract

Mealworms (*Tenebrio molitor* larva) are an edible insect and a protein-rich food; however, research on mealworms as a substitute protein is insufficient. In this study, mealworm fermentation extract (TMP) was assessed as a replacement for soy protein (SP) in a control diet (CON) or a high-fat diet (HFD) of mice for 12 weeks. TMP substitution reduced body weight, body weight gain, body fat mass (perirenal and mesenteric), fat size, glucose intolerance, and insulin resistance compared to the HFD-SP group. TMP alleviated hepatic steatosis (lipid contents and lipid droplets) in high-fat-fed mice and down-regulated the *PPARγ*, *CD36*, and *DGAT2* gene levels. Proteomic analysis showed that a HFD for 12 weeks up-regulated 20 proteins and down-regulated 17 proteins in mice fed SP. On the other hand, TMP reversed the protein profiles. TMP significantly down-regulated KHK, GLO1, ATP5H, SOD, and DDAH1 and up-regulated DLD, Mup1, CPS1, Ces3b, PDI, and HYOU1 compared to the HFD-SP group. These proteins are involved in the glucose, lipid, and amino acid metabolism, as well as in oxidative stress and endoplasmic reticulum stress. Thus, substituting SP for TMP helped improve HFD-induced obesity, steatosis, and insulin resistance in mice. These results suggest that TMP is a potential substitute for commonly used protein sources.

## 1. Introduction

The epidemic of obesity has increased globally and is consistently related to the rising prevalence of non-alcoholic fatty liver disease (NAFLD). NAFLD is distinguished by more than 5% fat in the liver and progresses to steatohepatitis, which can lead to fibrosis and ultimately to cirrhosis and hepatocellular carcinoma [1]. Many studies have indicated that the prevalence of NAFLD in obese individuals was higher than in lean individuals in the Asia-Pacific region, Netherlands, and the USA [2,3,4,5,6].

Edible insects have been utilized as alternative protein sources. Besides, their market size has increased continuously [7]. Previous studies have showed the hepatoprotective influences of insect extracts, such as *Oxya chinensis sinuosa*, *Gryllus bimaculatus*, and *Protaetia brevitarsis seulensis*, in high-fat diet (HFD)-induced NAFLD animals [8]. Among edible insects, *Tenebrio molitor* larvae are known as mealworms, which have a high quantity and quality of amino acid and protein [9]. Therefore, they are considered a highly sustainable protein source and alternative to soy protein (SP) or livestock meat [9]. Gessner et al. [10] reported that the isonitrogenous replacement of casein with mealworms exhibited lipid-lowering efficacy in the plasma and liver by inhibiting hepatic triglyceride (TG) and cholesterol synthesis in hyperlipidemic obese Zucker rats. Sim et al. [11] reported that fermented mealworm powder improved NAFLD in orotic acid-induced steatosis rats. A recent study reported that defatted mealworm fermentation extract (TMP) attenuated alcoholic hepatic steatosis in rats by regulating lipid synthesis, inflammation, and glutathione contents and changed the hepatic metabolites and gut microbiota [12]. TMP has a high percentage of free amino acids and essential amino acids compared to non-fermented mealworm extract.

Previous meta-analysis showed that plant protein intake was associated with a decreased risk of obesity, dyslipidemia, and diabetes, whereas animal proteins were associated with an increase [13,14]. Nutritional intervention research in animals and human beings suggests that SP reduces fat mass, body weight, and plasma cholesterol and TG levels [15]. Therefore, this study examined the effects of TMP as a substitute protein for SP in mice with obesity-induced hepatosteatogenesis and compared its underlying mechanisms.

## 2. Results

### 2.1. TMP Reduced the Body Weight and Fat Size

As expected, feeding a HFD for 12 weeks increased the body fat mass, fat size, and body weight in mice significantly compared with the control diet (CON)-fed mice (Figure 1). The fat size, body weight gain, and body weight showed a significant interaction between fat and TMP. HFD containing TMP (HFD-TMP) substitution effectively lessened the fat size, body weight gain, and body weight compared to the HFD containing soy protein (HFD-SP) group by 46%, 26%, and 13%, respectively (Figure 1). TMP did not affect food intakes (data not shown).

HFD-TMP significantly reduced mesenteric and perirenal weights of white adipose tissue (WAT) compared with HFD-SP. The weight of subcutaneous WAT was similar to the groups (Figure 1).

### 2.2. TMP Affected the Insulin Resistance Induced by High-Fat Diet

The blood glucose level did not differ between groups, whereas the serum insulin concentration and homeostasis model assessment of insulin resistance (HOMA-IR) showed only a TMP effect. Although there was no statistical significance, HFD-TMP lowered the insulin level compared to the HFD-SP, which showed a significant decrease in the HOMA-IR in the HFD-TMP group (Figure 2). The glucose tolerance test and serum leptin level also showed only a TMP effect. That is, HFD-TMP decreased glucose intolerance and leptin level significantly compared to the HFD-SP (Figure 2).

### 2.3. TMP Attenuated Hepatic Steatosis Induced by High-Fat Diet

Oil Red O and hematoxylin and eosin (H&E) staining indicated that the hepatic lipid droplets were higher in the HFD-SP group than in the CON-SP group. Trichrome staining also showed slight fibrosis progressing in the HFD groups. On the other hand, TMP reversed the histological changes (Figure 3). There were no significant differences in hepatic TG content between CON-SP and HFD-SP groups; however, HFD-TMP decreased the TG level significantly compared to the HFD-SP, by 23%. The hepatic free fatty acid (FFA) concentration was significantly lowered by TMP substitution in both the CON and HFD, while cholesterol level had no significant effect on TMP or fat, respectively (Figure 3).

TMP affected *PPARγ* and *CD36* genes, whereas fat affected *PPARγ* and *ChREBP* genes in the liver. *DGAT2* gene expression showed the significant interaction between TMP and fat. Therefore, HFD-TMP significantly down-regulated the lipogenic transcription factor *PPARγ* and TG synthesis gene *DGAT2* expression compared to the HFD-SP.

### 2.4. TMP Modified Hepatic Changes of Proteomic Profiles

To determine the effect of TMP on hepatic protein changes, proteomic analysis was carried out. When compared to the CON-SP, 20 proteins (ALDOB, ADH, KHK, FBP1, PC, GLO1, hsc71, HSPA9, BHMT, PHB, MAT1A, BCKDHA, hutU, MGCS2, ATP5H, SOD, DDAH1, GSTM2, ALDH1A1, and IMMT) were up-regulated and 17 proteins (ACO1, DLD, Mup1, CPS1, Ces3b, Ces3a, PDI, HSPA5, HYOU1, PNP, NME1, GSTP1, DDT, Tf, FGA, ALDH6A1, and ALDH12A1) were down-regulated in the HFD-SP group; and HFD-TMP significantly down-regulated KHK, GLO1, ATP5H, SOD, and DDAH1 and up-regulated DLD, Mup1, CPS1, Ces3b, PDI, and HYOU1 compared to the HFD-SP (Figure 4 and Figure 5). These proteins are related to the lipid metabolism, fructose/glucose metabolism, amino acid metabolism, and endoplasmic reticulum (ER) and oxidative stress (Table 1 and Table 2). When comparing CON-SP and CON-TMP, the pattern of changes was similar to the change between HFD-SP and HFD-TMP, but the change was less than that of the HFD (data not shown).

### 2.5. TMP Has a Similar Effect on the Serum Lipid Contents and Inflammation Markers to SP

Alanine aminotransferase (ALT) and aspartate aminotransferase (AST) activities, TG level, high-density lipoprotein (HDL)-cholesterol/total cholesterol ratio (HTR), atherogenic index (AI) in the serum, and fecal FFA level showed only a fat effect. The fecal TG and cholesterol concentration revealed an interaction between TMP and fat. TMP did not change the parameters significantly (Table 3).

## 3. Discussion

A recent study reported that defatted TMP (200 mg/kg body weight, oral supplementation) prevented alcoholic hepatic steatosis in rats by regulating *de novo* TG and cholesterol synthesis [12]. It had a higher amino acid content than the non-fermented extract. Moreover, its total and essential free amino acid levels were approximately 170 and 112 times higher than those of SP used in this study, respectively (data not shown). Therefore, this study examined the role of TMP as an alternative protein by comparing it with SP. SP is a beneficial plant protein for various metabolic diseases; for example, obesity, diabetes, and hyperlipidemia.

HFD-TMP fed for 12 weeks helped suppress the body weight gain, fat size, and visceral fat mass (perirenal and mesenteric WAT) in mice. Hepatic TG and FFA contents in HFD-SP were not significantly different compared to the CON-SP, but SP did not suppress the high-fat-induced lipid droplets, as evidenced by H&E and Oil Red O staining. However, HFD-TMP decreased hepatic lipid accumulation (lipid contents and lipid droplets) as compared to the HFD-SP group, which might be mediated by the down-regulation of *PPARγ* (lipid synthesis transcription factor) and its downstream genes, such as *CD36* (fatty acid uptake) and *DGAT2* (TG synthesis).

Proteomic analysis was accomplished the understanding of the underlying mechanism of the anti-steatotic effects of TMP. The proteins whose expression was increased or decreased by the HFD-SP were also affected by HFD-TMP. The expressed proteins were involved mainly in fructolysis/gluconeogenesis, energy expenditure, amino acid metabolism, and stress response.

The fructolysis/gluconeogenesis-controlling proteins, such as ALDOB, KHK, and FBP1, were up-regulated in the HFD-SP group compared with the CON-SP group, which is consistent with a previous result [16]. Fructose is metabolized mainly via ketohexokinase (KHK), which phosphorylates fructose to fructose-1-phosphate in the liver. ALDOB then splits fructose-1-phosphate into dihydroxyacetone phosphate (DHAP) and glyceraldehyde (GA). Fructose-derived triose-phosphate can be used for gluconeogenesis or *de novo* lipogenesis [17]. A previous study announced that the suppression of KHK in the liver resulted in a reduction of the enzymes involved in fatty acid synthesis [18]. The results showed that TMP substitution down-regulated KHK and FBP1 expression compared to the HFD-SP group; the change in ALDOB expression showed a lower trend. FBP1 is a gluconeogenesis enzyme that is promoted by obesity and high fat intake [19,20]. Therefore, TMP might suppress hepatic lipogenesis and HOMA-IR by regulating the fructose/glucose-related metabolism, such as KHK and FBP1, in HFD-fed mice.

Interestingly, an HFD-SP suppressed DLD and ACO1 protein expression, whereas it induced GLO1 protein expression compared to the CON-SP group. The decreases in DLD and ACO1 protein expression mean a decrease in the tricarboxyl acid (TCA) cycle, which can lead to an increase in methylglyoxal production from triose-phosphate. Methylglyoxal is the most potent glycating agent. Its accumulation in the liver could be an early incident in decreased insulin sensitivity due to binding to the major players of the insulin-signaling pathway [21]. GLO1 is involved in the detoxification of methylglyoxal that is mainly produced by removing phosphate from glyceraldehyde 3-phosphate (GA3P) and DHAP [21]. A previous proteomics study presented that the GLO1 protein in the liver of ob/ob mice was decreased threefold compared to the lean controls [22]. In the current study, GLO1 protein expression was up-regulated under the condition of HFD consumption, but TMP significantly suppressed GLO1, indicating lower triose-phosphate production in the HFD-TMP group. The above-mentioned reduction of DLD and ACO1 by the HFD may be associated with the up-regulation of ATP5H. ATP5H, a subunit of ATP synthase, produces ATP [23], which is involved in the adaptive response of SP on a HFD. However, this activated ATP5H is related to the production of reactive oxygen species (ROS). In this study, SOD and DDAH1 were up-regulated by the HFD. DDAH1 plays a major role in the degradation of asymmetrical dimethylarginine, which is an inhibitor of nitric oxide synthesis [24]. On the other hand, TMP reversed the changes in ATP5H, SOD, and DDAH1 induced by the HFD.

TMP dramatically up-regulated Mup1 expression that had been decreased by the HFD. Mup1 is a secreted protein produced predominantly in the liver, which increases energy expenditure and physical activity and improves the glucose intolerance and hyperinsulinemia [25]. Zhou et al. [26] suggested that hepatic Mup1 reduction contributes to hyperglycemia, insulin resistance, and glucose intolerance in diabetes. Baur et al. [27] reported that resveratrol increased mitochondrial biogenesis and energy expenditure in high-fat-induced obese mice, which was mediated by up-regulating hepatic Mup1 expression. CPS1 protein expression in the HFD-TMP was also up-regulated significantly compared to the HFD-SP. CPS1 is required for ammonia elimination and amino acid catabolism, whose gene and protein expression was reduced by the HFD [27]. De Chiara et al. [28] showed that hepatosteatosis was associated with decreased CPS1 expression in a dose-dependent manner [28]. NAFLD progression is related to the mitochondrial dysfunction and metabolites conversion of the urea cycle [29]. Previous studies reported that NAFLD patients [29] and a diet-induced non-alcoholic steatohepatitis (NASH) model [30] are deficient in ureagenesis. In this study, TMP up-regulated CPS1 accompanied by a decrease in fat in the liver.

A dysfunctional metabolism is located at the center of the pathogenesis for NAFLD and involves mitochondrial dysfunction, lipid dysmetabolism, and oxidative stress. Oxidative stress is believed to be a key etiology for NAFLD [31]. The ER provides an environment for disulfide formation and protein folding. Each disulfide bond generated during oxidative protein folding creates a single ROS. It is estimated that secretory cells produce 3–6 million disulfide bonds per minute. In this way, protein folding in the ER is linked to the production of ROS and oxidative stress [32,33].

On the contrary, cellular oxidative stress can interfere with ER homeostasis and induce ER stress [34,35,36]. In the current study, a HFD-SP caused the down-regulation of PDI, HSPA5, and HYOU1 compared to the CON-SP group, but HFD-TMP up-regulated HYOU1 and PDI significantly relative to the HFD-SP. HYOU1, an ER chaperone protein, increases in various tissues under hypoxic conditions but is highly expressed in the pancreatic β cells and liver, even in normoxia [37,38]. High expression of HYOU1 is considered to protect ER stress and can lower insulin resistance in mice [39,40]. Wang et al. [41] reported that the amelioration of lipid-induced ER stress in the liver by Sirt1/AMPK pathway activation is associated with the induction of HYOU1. Moreover, protein disulfide-isomerase (PDI) is required for optimal insulin production to maintain glucose homeostasis in metabolic diseases [42]. The results of the present study suggest that the selective up-regulation of protein chaperones by TMP alternative supplementation in the liver should improve insulin action and hepatic steatosis.

Finally, Ces3b hydrolyzes long-chain fatty acids and thioesters that would play a role in the lipid metabolism. Ces3b helps provide substrates for the assembly of very-low-density lipoprotein (VLDL) in the liver. Previous in vitro and in vivo studies showed that the suppression of Ces3b lowered VLDL secretion [43]. Blocking VLDL assembly and secretion can lead to severe lipid accumulation in the liver [44]. This situation was avoided in TG hydrolase-deficient mice because of the compensatory effect of the decreased non-esterified fatty acid flux from the adipose tissue to the liver and the elevated fatty acid oxidation [45,46]. Elevated TGH expression was observed in patients with steatosis and NASH [43]. This study showed that down-regulation of Ces3b in the HFD-TMP group might be related to lower hepatic lipid concentration.

In conclusion, the substitution of SP with TMP alleviated HFD-induced obesity, hepatosteatosis, and insulin resistance, which might be mediated by regulating the protein expression related to energy expenditure, lipid synthesis from glucose, and ER stress in the liver of mice. These results propose that TMP is possibly better than SP under HFD feeding conditions. Moreover, TMP may potentially be a substitute for a commonly used protein source.

## 4. Materials and Methods

### 4.1. TMP Preparation

Preparation of TMP was as described in a previous study [12]. Defatted and freeze-dried powder was fermented for 72 h using a *Saccharomyces cerevisiae* strain (KCTC 17299, Korean Cell Line Bank, Seoul, Korea). Fermented mealworm powder was added to replace the peptone in the yeast–peptone dextrose medium. After that, 1 L yeast/defatted mealworm fermented broth were with 70% fermented alcohol (1L). Extracts were concentrated and freeze-dried.

### 4.2. Animal Experiments

Male C57BL/6N mice (4-week-old, *n* = 40) were obtained (Orient Bio Inc., Seoul, Korea) and housed under controlled light (light 8 a.m.–8 p.m., dark 8 p.m.–8 a.m.), temperature (22 ± 2 °C), and humidity (50 ± 5%) conditions. Mice had free access to food and water. Experiments were performed in accordance with the Sunchon National University Institutional Animal Care and Use Committee ethical guidelines (approved under no. SCNU_IACUC-2020–08).

After 7 days of adaptation, the mice were divided randomly into the four groups (n = 10 per group): control diet containing SP (CON-SP), control diet containing TMP (CON-TMP), high-fat diet containing SP (HFD-SP), and high-fat diet containing TMP (HFD-TMP). All groups received an AIN-76 standard diet with 20% protein. SP (Shandong Yuxin Bio-Tech Co., Ltd., Shandong, China) and TMP groups had 20% SP or 20% TMP added into the control and high-fat diet, respectively, for 12 weeks. Composition of the diets is presented in Table 4. Food intake and body weight were monitored three times and once per week, respectively. At the end of the experimental period, mice were fasted overnight and anesthetized. The organs (serum, liver, and WAT) were collected.

### 4.3. Oral Glucose Tolerance Test

After 12 weeks of treatment, the mice were fasted for 6 h, and then administered glucose (1 g/kg body weight). Blood samples were collected from the tail vein at four time-points: 0, 30, 60, and 120 min after oral glucose load. The fasting blood glucose concentration was monitored using a G-doctor (AllMedicus, Co., Ltd., Anyang, Korea).

### 4.4. Serum Parameters

The serum insulin (M1104, Morinaga Institute of Biological Science, Inc., Yokohama, Japan), tumor necrosis factor-α (TNF-α; BMS607HS, Invitrogen, Carlsbad, CA, USA), interleukin-6 (IL-6; BMS603HS, Invitrogen), adiponectin (DY1119, R&D Systems, Minneapolis, MN, USA), and leptin levels (DY498, R&D Systems) were determined using an enzyme-linked immunosorbent assays (ELISA) kit. The serum AST and ALT levels were measured using a Fuji Dri-Chem 3500i (Fujifilm, Tokyo, Japan). The serum glucose level was determined using the enzymatic method (Asan Pharmaceutical Co., Ltd., Seoul, Korea). The HOMA-IR was calculated using the following formula: insulin (µIU/mL) × glucose (mmol/L) / 22.5.

### 4.5. Lipid Profile Evaluation

The serum TG, cholesterol, HDL cholesterol (Asan Pharmaceutical Co., Ltd.), and FFA (Shinyang Diagnostics, Seoul, Korea) levels were measured using reagents according to the manufacturers’ instructions. The hepatic and fecal lipid contents (TG, FFA, and cholesterol) were monitored after extraction, according to Folch et al. [47].

### 4.6. Histological Analysis

The liver and epididymal WAT were fixed in 10% neutral buffered formalin and stained by H&E. The liver sections were stained with trichrome and Oil Red O. All images were captured using an optical microscope at 200× magnification.

### 4.7. RNA Isolation and Quantitative Real-Time PCR Analysis

The total RNA was isolated from the liver tissue using a TRIzol reagent (15596018, Invitrogen) according to the manufacturer’s instructions. The RNA concentration and purity were evaluated using a Nanodrop 2000 spectrophotometer (Thermo Fisher Scientific, Waltham, MA, USA). The complementary DNA (cDNA) was synthesized from the total RNA (1 μg) using a ReverTra Ace qPCR RT master mix (FSQ-201, Toyobo, Osaka, Japan). Real-time PCR was performed using an SYBR green PCR kit (204143, Qiagen, Hilden, Germany) with a specific primer (Appendix A) in a CFX96 Touch^TM^ real-time PCR detection system (Bio-Rad Laboratories, Inc., Hercules, CA, USA). The data were analyzed using the 2^−△△CT^ method [48] and normalized to *GAPDH* in the same cDNA samples.

### 4.8. Protein Sample Preparation

The liver tissues were homogenized directly using motor driven homogenizer (PowerGen125, Fisher Scientific, Pittsburgh, PA, USA) in sample lysis solution containing 7 M urea (Sigma-Aldrich, St. Louis, MO, USA), 2 M thiourea (Sigma-Aldrich) containing 4% (*w*/*v*) 3-[(3-cholamidopropy)dimethyammonio]-1-propanesulfonate (CHAPS, Sigma-Aldrich), 1% (*w*/*v*) dithiothreitol (DTT, Sigma-Aldrich), 2% (*v*/*v*) pharmalyte (Amersham Biosciences, Little Chalfont, UK), and 1 mM benzamidine (Sigma-Aldrich). The proteins were extracted for one hour at room temperature with vortexing. After centrifugation at 15,000× *g* for one hour at 15 °C, the insoluble material was discarded and the soluble fraction was used for two-dimensional gel electrophoresis. The protein concentration was assayed using the Bradford method [49].

### 4.9. Two-Dimensional Polyacrylamide Gel Electrophoresis (2-DE)

2-DE was conducted as previously described [50]. For 2-DE analysis, immobilized pH gradient (IPG) strips (Genomine Inc., Pohang, Korea) were rehydrated in swelling buffer containing 7 M urea (Sigma-Aldrich), 2 M thiourea (Sigma-Aldrich), 1% (*w*/*v*) DTT (Sigma-Aldrich), and 4% (*w*/*v*) CHAPS (Sigma-Aldrich). Isoelectric focusing (IEF) was performed at 20 °C. The gels were stained with Coomassie G250, and then destained using deionized water and images were acquired with an image scanner (Bio-Rad Laboratories) as described by Anderson et al. [31].

### 4.10. Peptide Mass Fingerprinting (PMF)

Peptide mass fingerprinting was performed as previous described [51]. For protein identification by peptide mass fingerprinting, the protein spots were excised, digested with trypsin (Promega, Madison, WI, USA), mixed with α-cyano-4-hydroxycinnamic acid (Sigma-Aldrich) in 50% acetonitrile (Sigma-Aldrich) with 0.1% trifluoroacetic acid (TFA, Sigma-Aldrich), and subjected to matrix-assisted laser desorption ionization-time of flight mass spectrometry (MALDI-TOF, Microflex LRF 20, Bruker Daltonics, Billerica, MA, USA), as described by Fernandez et al. [52]. The spectra were collected from 300 shots per spectrum over the *m*/*z* range 700–4000 and calibrated by two-point internal calibration using Trypsin auto-digestion peaks (*m*/*z* 842.5099, 2211.1046). The peak list was generated using Flex Analysis 3.0. The search program MASCOT, developed by The Matrixscience (http://www.matrixscience.com/, accessed on 8 December 2020), was used for protein identification by peptide mass fingerprinting.

### 4.11. Statistical Analysis

The data are presented as the means ± standard error (S.E.), and *p* values < 0.05 were considered significant. Statistical analysis was performed using the Statistical Package for the Social Sciences (SPSS version 26, SPSS Inc., Chicago, IL, USA). The data were analyzed by two-way ANOVA followed by a Turkey post hoc test to examine the differences between the group means. The proteomic data between the HFD-SP group and HFD-TMP group were compared using a Student’s *t*-test.

## Figures and Tables

**Figure 1 ijms-22-03615-f001:**
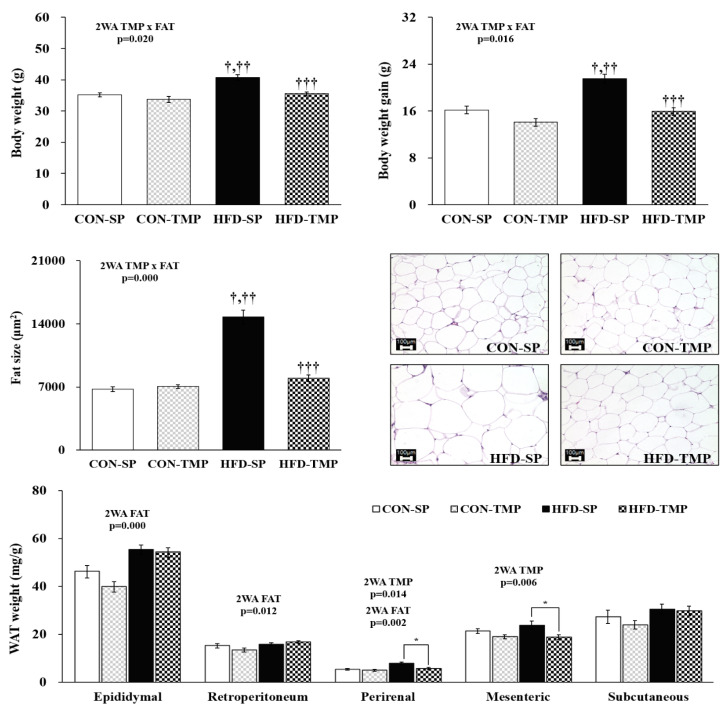
Body weight, body weight gain, fat mass, and adipocyte size. The values are expressed as the means ± S.E. (n = 10 per group). ^†^ vs. CON-SP, ^††^ vs. CON-TMP and ^†††^ vs. HFD-SP by two-way ANOVA followed by a Turkey post hoc test (*p* < 0.05). * *p* < 0.05 vs. HFD-SP by Student’s *t*-test. Histological analysis magnification 200×. 2WA FAT, fat diet effect in two-way ANOVA (*p* < 0.05); 2WA TMP, TMP effect in two-way ANOVA (*p* < 0.05); 2WA TMP × FAT, interaction between TMP and fat diet in two-way ANOVA (*p* < 0.05); CON-SP, control diet containing SP; CON-TMP, control diet containing TMP; HFD-SP, high-fat diet containing SP; HFD-TMP, high-fat diet containing TMP.

**Figure 2 ijms-22-03615-f002:**
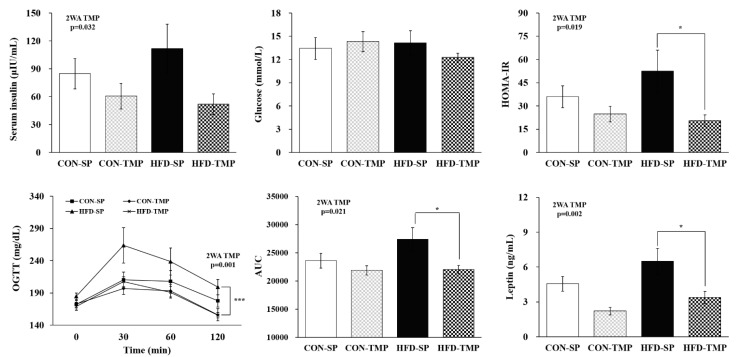
Serum insulin, glucose, and leptin levels; HOMA-IR; and OGTT. The values are expressed as the means ± S.E. (n = 10 per group). * *p* < 0.05, *** *p* < 0.001 vs. HFD-SP by Student’s *t*-test. 2WA FAT, fat diet effect in two-way ANOVA (*p* < 0.05); 2WA TMP, TMP effect in two-way ANOVA (*p* < 0.05); CON-SP, control diet containing SP; CON-TMP, control diet containing TMP; HFD-SP, high-fat diet containing SP; HFD-TMP, high-fat diet containing TMP; HOMA-IR, homeostasis model assessment of insulin resistance; OGTT, oral glucose tolerance test.

**Figure 3 ijms-22-03615-f003:**
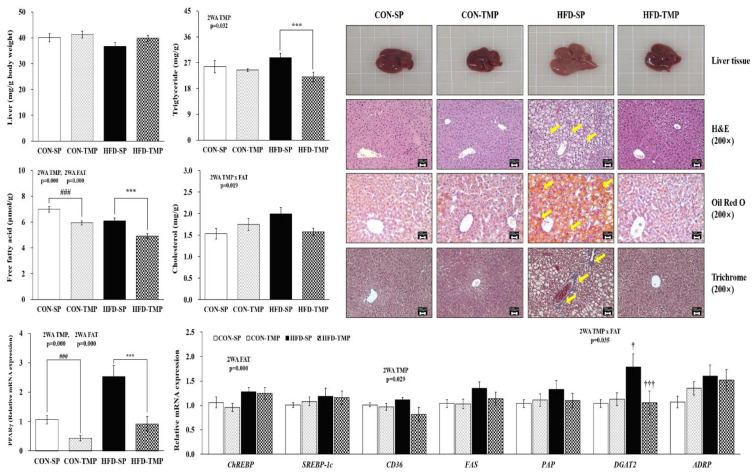
Histology, lipid contents, and lipid metabolism-related gene expression in the liver. The values are expressed as the means ± S.E. (n = 10 per group). ^†^ vs. CON-SP and ^†††^ vs. HFD-SP by two-way ANOVA followed by a Turkey post hoc test (*p* < 0.05). ### *p* < 0.001 vs. CON-SP, *** *p* < 0.001 vs. HFD-SP by Student’s *t*-test. Histological analysis magnification 200×. Yellow arrows indicated the lipid droplets or fibrosis. 2WA FAT, fat diet effect in two-way ANOVA (*p* < 0.05); 2WA TMP, TMP effect in two-way ANOVA (*p* < 0.05); 2WA TMP × FAT, interaction between TMP and fat diet in two-way ANOVA (*p* < 0.05); CON-SP, control diet containing SP; CON-TMP, control diet containing TMP; HFD-SP, high-fat diet containing SP; HFD-TMP, high-fat diet containing TMP; H&E, hematoxylin and eosin.

**Figure 4 ijms-22-03615-f004:**
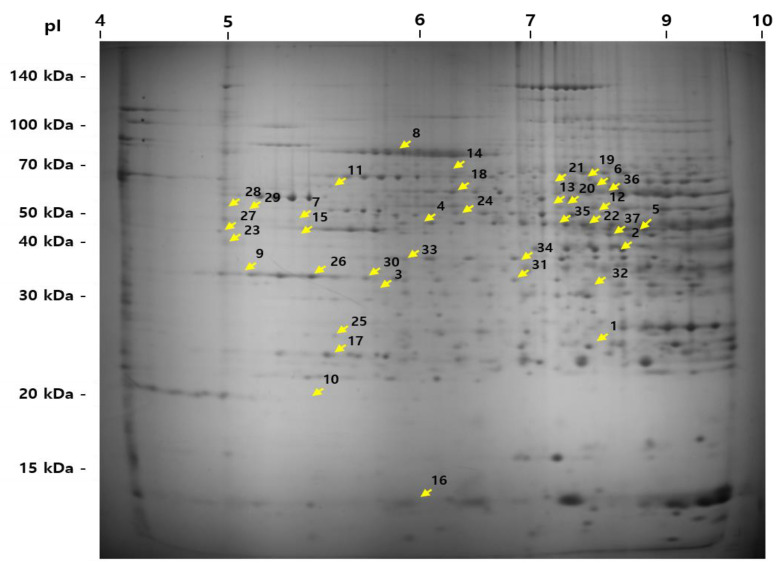
Two-dimensional electrophoresis of proteins in the liver. The 2-D PAGE image from the liver was used as a master gel and reference map. The HFD caused 37 spots to change. The protein spots were identified by MALDI-TOF (arrow) and are marked by their spot numbers. pI, isoelectric point.

**Figure 5 ijms-22-03615-f005:**
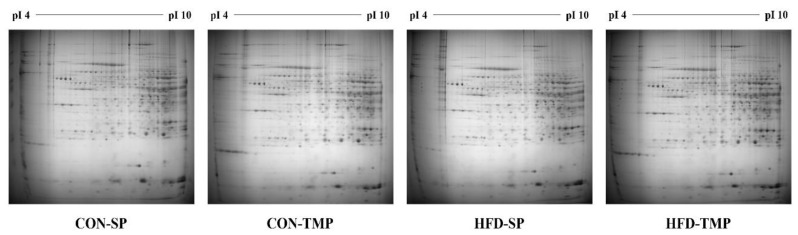
Two-dimensional electrophoresis patterns of the proteins in the liver.

**Table 1 ijms-22-03615-t001:** Effect of TMP on the proteins up-regulated by the high-fat diet.

No.	Symbol	Protein Description	Accession	Fold Change Protein Ratio			Protein Sequence Coverage (%) ^b^	MW	pI	Score ^c^
				B/A	C/B	*p*-Value of C/B ^a^				
1	ALDOB	Fructose-bisphosphate aldolase B	NP_659152.1	1.9	0.05	0.306	45	24	8.52	148
2	ADH	Alcohol dehydrogenase 1	NP_031435.1	2.38	0.17	0.335	32	39	8.44	114
3	KHK	Ketohexokinase	NP_032465.2	1.57	0.62	0.019	63	30	5.81	158
4	FBP1	Fructose-1,6-bisphosphatase 1	NP_062268.1	1.6	0.59	0.057	61	46	6.15	242
5	PC	Pyruvate carboxylase, mitochondrial	NP_000911.2	1.69	0.61	0.404	46	62	6.25	447
6	GLO1	Lactoylglutathione lyase (=glyoxalase 1)	NP_079650.3	1.44	0.43	0.029	72	38	5.24	182
7	hsc71	Heat shock cognate 71 kDa	NP_112442.2	3.23	0.39	0.311	44	46	5.37	224
8	HSPA9	Stress-70 protein	NP_034611.2	2.73	0.26	0.277	37	72	5.81	274
9	BHMT	Betaine—homocysteine S-methyltransferase 1	NP_057877.1	2.78	0.07	0.351	57	56	8.01	160
10	PHB	Prohibitin	NP_032857.1	1.82	0.22	0.138	53	20	5.57	181
11	MAT1A	S-adenosylmethionine synthase isoform type-1	NP_598414.1	1.6	0.47	0.089	48	53	5.51	183
12	BCKDHA	2-oxoisovalerate dehydrogenase subunit alpha	NP_031559.3	1.91	0.63	0.117	48	45	8.15	160
13	hutU	Urocanate hydratase	NP_659189.2	1.47	0.62	0.275	41	44	7.27	290
14	HMGCS2	Hydroxymethylglutaryl-CoA synthase, mitochondrial	NP_032282.2	1.65	0.1	0.281	37	42	8.65	120
15	ATP5H	ATP synthase subunit d, mitochondrial	NP_082138.1	1.69	0.68	0.035	65	49	5.52	108
16	SOD	Superoxide dismutase	NP_035564.1	1.93	0.42	0.022	42	10	6.02	108
17	DDAH1	N(G),N(G)-dimethylarginine dimethylaminohydrolase 1	NP_081269.1	1.57	0.18	0.022	35	25	5.64	88
18	GSTM2	Glutathione S-transferase Mu 7	NP_080948.2	1.79	0.6	0.542	49	56	6.34	146
19	ALDH1A1	Retinal dehydrogenase 1	NP_038495.2	1.46	0.21	0.299	37	63	7.92	123
20	IMMT	MICOS complex subunit MIC60	NP_001240617.1	1.78	0.56	0.631	33	44	7.65	135

^a^ The *p*-value means Student’s *t*-test between the B and C groups. ^b^ Protein sequence coverage (%) is defined as the percentage of the whole length of the protein sequence which is covered by matched peptides identified by the MALDI-TOF/MS analysis. ^c^ Mascot scores greater than 61 in NCBI *p* ≤ 0.05 from Mascot search on MALDI-TOF/MS data were considered. A, control diet containing SP (CON-SP); B, high-fat diet containing SP (HFD-SP); C, high-fat diet containing TMP (HFD-TMP).

**Table 2 ijms-22-03615-t002:** Effects of TMP on the proteins down-regulated by the high-fat diet.

No.	Symbol	Protein Description	Accession	Fold Change Protein Ratio			Protein Sequence Coverage (%) ^b^	MW	pI	Score ^c^
				B/A	C/B	*p*-Value of C/B ^a^				
21	ACO1	Cytoplasmic aconitate hydratase	NP_031412.2	0.49	2.03	0.141	48	56	7.23	311
22	DLD	Dihydrolipoyl dehydrogenase, mitochondrial	NP_031887.2	0.43	5.62	0.027	52	45	7.99	172
23	Mup1	Major urinary protein 1	NP_001334083.1	0.49	18.3	0.034	66	34	4.96	130
24	CPS1	Carbamoyl-phosphate synthase ammonia, mitochondrial	NP_001074278.1	0.29	7.26	0.019	28	51	6.48	249
25	Ces3b	Carboxylesterase 3B	NP_001152887.1	0.34	3.49	0.000	20	28	5.65	84
26	Ces3a	Isoform 2 of carboxylesterase 3A	NP_001158153.1	0.46	2.73	0.069	34	35	5.43	123
27	PDI	Protein disulfide-isomerase	NP_035162.1	0.62	2.47	0.043	50	42	4.77	223
28	HSPA5	Endoplasmic reticulum chaperone BiP	NP_001156906.1	0.68	1.31	0.611	41	53	5.07	268
29	HYOU1	Hypoxia up-regulated protein 1	NP_067370.3	0.29	7.26	0.019	22	51	5.12	78
30	PNP	Purine nucleoside phosphorylase	NP_038660.1	0.67	1.41	0.321	41	31	5.78	99
31	NME1	Nucleoside diphosphate kinase A	NP_032730.1	0.59	1.58	0.107	73	33	6.84	131
32	GSTP1	Glutathione S-transferase P 1	NP_038569.1	0.54	1.6	0.236	62	30	7.68	164
33	DDT	D-dopachrome decarboxylase	NP_034157.1	0.29	1.71	0.429	85	35	6.09	107
34	Tf	Serotransferrin	NP_598738.1	0.61	1.43	0.217	33	36	6.94	156
35	FGA	Isoform 2 of fibrinogen alpha chain	NP_001104518.1	0.62	2.95	0.272	33	47	7.16	122
36	ALDH6A1	Methylmalonate-semialdehyde dehydrogenase (acylating), mitochondrial	NP_598803.1	0.61	3.79	0.217	25	57	8.29	67
37	ALDH12A1	Delta-1-pyrroline-5-carboxylate dehydrogenase, mitochondrial	NP_780647.3	0.62	2.95	0.128	26	47	8.45	80

^a^ The *p*-value means Student’s *t*-test between the B and C groups. ^b^ Protein sequence coverage (%) is defined as the percentage of the whole length of the protein sequence which is covered by matched peptides identified by the MALDI-TOF/MS analysis. ^c^ Mascot scores greater than 61 in NCBI *p* ≤ 0.05 from Mascot search on MALDI-TOF/MS data were considered. A, control diet containing SP (CON-SP); B, high-fat diet containing SP (HFD-SP); C, high-fat diet containing TMP (HFD-TMP).

**Table 3 ijms-22-03615-t003:** Serum markers and fecal lipid contents.

	CON-SP	CON-TMP	HFD-SP	HFD-TMP	2WA ^1^
**Serum markers**					
**AST (U/L)**	49.55 ± 3.33	39.33 ± 1.56	53.00 ± 5.29	42.77 ± 2.67	F ^2^ (*p* = 0.008)
**ALT (U/L)**	22.36 ± 1.95	17.22 ± 3.48	33.82 ± 5.57	27.80 ± 6.22	F (*p* = 0.018)
**TG (mg/dL)**	97.01 ± 9.79	74.84 ± 8.85	78.15 ± 7.15	70.38 ± 6.51	F (*p* = 0.000)
**FFA (μmol/L)**	0.64 ± 0.04	0.55 ± 0.04	0.52 ± 0.04	0.52 ± 0.04	
**TC (mg/dL)**	120.01 ± 5.40	112.91 ± 5.51	149.41 ± 8.67	153.84 ± 6.58	
**HDL-C (mg/dL)**	84.64 ± 4.72	87.52 ± 4.56	92.47 ± 5.32	97.03 ± 5.64	
**HTR (%)**	70.40 ± 2.59	77.53 ± 1.87	62.19 ± 2.17	63.34 ± 3.25	F (*p* = 0.000)
**AI**	0.44 ± 0.05	0.30 ± 0.03	0.63 ± 0.05	0.62 ± 0.08	F (*p* = 0.000)
**TNF-α (pg/mL)**	9.84 ± 0.71	9.49 ± 0.80	10.68 ± 0.91	10.86 ± 0.85	
**IL-6 (pg/mL)**	18.07 ± 4.67	11.84 ± 1.21	17.63 ± 2.69	16.51 ± 2.54	
**Adiponectin (μg/mL)**	3.63 ± 0.18	3.55 ± 0.16	3.60 ± 0.14	3.36 ± 0.18	
**Fecal lipid contents**					
**TG (mg/g)**	67.94 ± 5.66	390.60 ± 15.67 ^†^	333.32 ± 15.26 ^†^	416.33 ± 73.37 ^†^	TxF ^3^ (*p* = 0.000)
**FFA (μmol/g)**	61.16 ± 4.63	64.83 ± 4.18	155.48 ± 7.75	172.52 ± 17.41	F (*p* = 0.000)
**Cholesterol (mg/g)**	66.39 ± 3.07	48.69 ± 1.54 ^†^	51.22 ± 2.49 ^†^	60.23 ± 3.50	TxF (*p* = 0.013)

The values are expressed as the means ± S.E. (n = 10 per group). ^†^ vs. CON-SP by two-way ANOVA followed by a Turkey post hoc test (*p* < 0.05). ^1^ 2WA, two-way ANOVA, ^2^ F, fat diet effect in two-way ANOVA (*p* < 0.05), ^3^ TxF, interaction between TMP and fat diet in two-way ANOVA (*p* < 0.05). CON-SP, control diet containing SP; CON-TMP, control diet containing TMP; HFD-SP, high-fat diet containing SP; HFD-TMP, high-fat diet containing TMP.

**Table 4 ijms-22-03615-t004:** Composition of experimental diets.

Ingredients (g/kg Diet)	CON-SP	CON-TMP	HFD-SP	HFD-TMP
Soy protein	200	-	200	-
TMP ^1^	-	200	-	200
DL-methionine	3	3	3	3
Choline bitartrate	2	2	2	2
Corn starch	500	500	340	340
Sucrose	150	150	150	150
Cellulose	50	50	50	50
Corn oil	50	50	30	30
Lard	0	0	180	180
Mineral mixture	35	35	35	35
Vitamin mixture	10	10	10	10

^1^*Tenebrio molitor* larva fermentation extract.

## Data Availability

The datasets used and/or analyzed during the current study are available from the corresponding author on reasonable request.

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
