# Peer review of "Effects of Edible Insect Tenebrio molitor Larva Fermentation Extract as a Substitute Protein on Hepatosteatogenesis and Proteomic Changes in Obese Mice Induced by High-Fat Diet"

_ijms, 2021, doi:10.3390/ijms22073615_

Round 1
Reviewer 1 Report
The aim of this study was to assess mealworm fermented extract (TMP) as a replacement for soy protein (SP) in a normal diet (ND) or a high-fat diet (HF). The key point is the replacement of one protein source for another.
The methods used are conducted properly and seems appropriate for this study. It is however reason to question the design of the study. When only two sources of proteins are compared , what is the proper control? Can the observed effects with TMP as the protein source be considered to be beneficial, or should the effects observed with SP be considered adverse? From a dietary point of view comparing diets in which on single protein source is completely replaced by another singel protein source is of limited interest. Studies in which a mix of proteins are partly replaced by a specific protein would have been more relevant.
The problems with comparing only two specific proteins are illustrated by the following examples
Sect 2.3. “Although soy protein prevented the significant increase in the hepatic triglyceride content by the high-fat diet;” Which high fat diets are compared?
The reference to a high fat diet is often inconsistent when a reference to the protein source is omitted.
No reference should be made to a HF diet without also stating the protein source.
Page 10 “As expected, soy protein prevented the significant increases in hepatic triglyceride and free fatty acid contents”. Prevention compared to which diet?
This is also illustrated by Table 1 and 2 showing TMP effects on upregulated and downregulated protein by a high fat diet. Is this compared with a ND diets containing TMP? If so, could a similar comparison have been made for diets with SP?
In presenting results and in the discussion one should be very aware of the fact that this study is a comparison of two different protein diets and not the result of adding a substance to a diet. Statement that TMP reverses (and improves) many of the effects seen compared with SP diet should be avoided as it indicates that a protein source has been added (not replacing a protein) to a control diet.
The term “interaction between TMP and fat” is used repeatedly which is confusing. For ex
Sect 2.3 “The hepatic free fatty acid concentration revealed the effects of TMP and fats, while cholesterol level showed the significant interaction between TMP and fat.” The difference between the terms “revealed the effects of “ and “significant interaction between” should be explained.
In table 1 and 2 the column headings to the right: “C/A D/C - Protein sequence coverage (%) - score” needs to be explained.
Irrelevant columns which are not discussed should be omitted.
Statistics.
In figures 1 – 3 the following notifications are used above the figures: 2WA FAT 2WA TMP 2WA TMPxFAT
These terms needs an explanation. Which groups are compared?
Is two way ANOVA an appropriate method?. Which two factors are included in the analysis?
Minor points
Fig 3, Should the unit on the vertical axes (LIVER) be mg/g body weight?
Table 4 legend “ Experimental diets were adjusted form considering of protein contents.” Please rewrite this sentence to make it understandable
Author Response
Revisions and statement made according to
Reviewer1 response and comments
Thank you very much for your kind review and valuable suggestions. We have modified the manuscript accordingly, and detailed corrections are listed below point by point:
- The methods used are conducted properly and seems appropriate for this study. It is however reason to question the design of the study. When only two sources of proteins are compared, what is the proper control?
Answer: In this study, soy protein was used as a control protein. The purpose of our research is to evaluate the possibility of TMP as an alternative dietary protein. Soy protein is already known to be effective against metabolic diseases such as obesity, diabetes, and dyslipidemia; however, it has a low essential amino acids contents. We previously confirmed that phenotype (i.e. body weight, fat size, and serum levels of glucose, insulin and leptin) of obese mice improved as the amount of TMP substituted for soy protein increased. Therefore, we tried to compare the effects of TMP and soy protein in lean and obese mice.
- Can the observed effects with TMP as the protein source be considered to be beneficial, or should the effects observed with SP be considered adverse?
Answer: As explained earlier, we suggest that TMP is better than soy protein, which is known as a good protein in obesity. The title has been revised to avoid confusion interpretation of the research purpose. “Effects of edible insect Tenebrio molitor larva fermentation extract as a substitute protein on hepatosteatogenesis and proteomic changes in obese mice induced by high-fat diet”
In addition, we revised the sentences entirely.
- Sect 2.1, line 2: The fat size, body weight gain, and body weight showed a significant interaction be-tween fat and TMP. TMP (HD+TMP) substitution effectively lessened the fat size, body weight gain, and body weight compared to the HD containing soy protein (HD+SP) group by 13%, 26%, and 46%, respectively (Fig. 1).
- Sect 2.1, line 7: HD+TMP significantly reduced mesenteric and perirenal weights of white adipose tissue (WAT) compared with HD+SP. The weight of subcutaneous WAT was similar to the groups (Fig. 1).
- Sect 2.2, line 3: Although there was no statistical significance, HD+TMP lowered the insulin level compared to the HD+SP, which showed a significant decrease in the HOMA-IR in the HD containing TMP (HD +TMP) group (Fig. 2). The glucose tolerance test and serum leptin level also showed only a TMP effect. That is, HD+TMP decreased glucose intolerance and leptin level significantly compared to the HD+SP (Fig. 2).
- Sect 2.3, line 4: There was no significant differences hepatic TG content between ND+SP and HD+SP groups; however, HD+TMP decreased the TG level significantly compared to the HD+SP by 23%.
- Sect 2.3, line 10: DGAT2 gene expression showed the significant interaction between TMP and fat. Therefore, HD+TMP significantly down-regulated the lipogenic transcription factor PPAR γ and TG synthesis gene DGAT2 expression compared to the HD+SP.
- From a dietary point of view comparing diets in which on single protein source is completely replaced by another single protein source is of limited interest. Studies in which a mix of proteins are partly replaced by a specific protein would have been more relevant.
Answer: As mentioned in answer #1, we evaluated the beneficial function of the dietary protein using TMP, and the control was used soy protein. To clarify the purpose of our research, we revised the titles and sentences. Please note that many researchers are comparing different types of proteins, such as casein, soy protein, whey protein, etc.
- Sect 2.3. “Although soy protein prevented the significant increase in the hepatic triglyceride content by the high-fat diet;” Which high fat diets are compared?
à Answer: Thank you for your kind point. Sect 2.3, line 4-6: We revised the sentence to “There was no significant differences hepatic TG content between ND+SP and HD+SP groups; however, HD+TMP decreased the TG level significantly compared to the HD+SP by 23%.”
- The reference to a high fat diet is often inconsistent when a reference to the protein source is omitted. No reference should be made to a HF diet without also stating the protein source.
Answer: According to your comment, we have addressed the reference the high-fat diet.
- Sect 2.2, line 3: HD+TMP lowered the insulin level compared to the HD+SP.
- Sect 2.4, line 2: When compared to the ND+SP, 20 proteins (ALDOB, ADH, KHK, FBP1, PC, GLO1, hsc71, HSPA9, BHMT, PHB, MAT1A, BCKDHA, hutU, MGCS2, ATP5H, SOD, DDAH1, GSTM2, ALDH1A1, and IMMT) were up-regulated and 17 proteins (ACO1, DLD, Mup1, CPS1, Ces3b, Ces3a, PDI, HSPA5, HYOU1, PNP, NME1, GSTP1, DDT, Tf, FGA, ALDH6A1, and ALDH12A1) were down-regulated in the HD+SP group, HD+TMP significantly down-regulated KHK, GLO1, ATP5H, SOD, and DDAH1 and up-regulated DLD, Mup1, CPS1, Ces3b, PDI, and HYOU1 compared to the HD+SP (Fig. 4 and 5).
- Discussion, line 13: “HD+TMP decreased hepatic lipid—”
- Discussion, line 18: “The proteins whose expression was increased or decreased by the HD+SP were modified proteins by HD+TMP.”
- Page 10 “As expected, soy protein prevented the significant increases in hepatic triglyceride and free fatty acid contents”. Prevention compared to which diet?
Answer: Sorry for the unclear expression. Discussion, line 10-11; we revised to “Hepatic TG and FFA contents in HD+SP were not significantly different compared to the ND+SP”
- This is also illustrated by Table 1 and 2 showing TMP effects on upregulated and downregulated protein by a high fat diet. Is this compared with a ND diets containing TMP? If so, could a similar comparison have been made for diets with SP?
Answer: We inserted one sentence in Sect 2.4, line 10-12. “When comparing ND+SP and ND+TMP, the pattern of changes was similar to change between HD+SP and HD+TMP, but the change was less than that of the HD (data not shown).
- In presenting results and in the discussion one should be very aware of the fact that this study is a comparison of two different protein diets and not the result of adding a substance to a diet. Statement that TMP reverses (and improves) many of the effects seen compared with SP diet should be avoided as it indicates that a protein source has been added (not replacing a protein) to a control diet.
Answer: As you suggested, we revised some sentences as follows;
- Subtitle 2.2. TMP improved à TMP affected
- Subtitle 2.4. TMP reversed hepatic changes of proteomic profiles by a high-fat diet à TMP modified hepatic changes of proteomic profiles
- For the rest, thanks for referring to answer #5.
- The term “interaction between TMP and fat” is used repeatedly which is confusing. For ex
Sect 2.3 “The hepatic free fatty acid concentration revealed the effects of TMP and fats, while cholesterol level showed the significant interaction between TMP and fat.” The difference between the terms “revealed the effects of “and “significant interaction between” should be explained.
Answer: According to your comment, we delete the sentences and rewrite the results. Sect 2.3, line 6: “The hepatic free fatty acid (FFA) concentration was significantly lowered in both the ND and HD, while cholesterol level had no significant effect on fat and TMP, respectively (Fig. 3).”
- In table 1 and 2 the column headings to the right: “C/A D/C - Protein sequence coverage (%) - score” needs to be explained.
Irrelevant columns which are not discussed should be omitted.
Answer: In proteomic analysis results, it is common to present factors related gene characteristics in the table. All factors did not need to be discussed. We have inserted sentences about what you pointed out in the table 1 and 2. A, normal diet containing SP (ND+SP); B, high-fat diet containing SP (HD+SP); C, high-fat diet containing TMP (HD+TMP). b) Protein sequence coverage (%) is defined as the percentage of the whole length of the protein sequence which is covered by matched peptides identified by the MALDI-TOF MS analysis. c) Mascot scores greater than 61 in NCBI and p ≤ 0.05 from Mascot search on MALDI-TOF/MS data were considered.
Statistics.
- In figures 1 – 3 the following notifications are used above the figures: 2WA FAT 2WA TMP 2WA TMPxFAT
These terms needs an explanation. Which groups are compared?
Answer: The terms used in Fig 1-3 are commonly presented in two-way ANOVA analysis results. Three null hypotheses: “There is no difference in group means at any level of the first independent variable (fat).”, “There is no difference in group means at any level of the second independent variable (TMP).”, and “The effect of one independent variable (fat) does not depend on the effect of the other independent variable (TMP) (a.k.a. no interaction effect).” A two-way ANOVA without interaction analyzes the hypothesis for each independent variable. We have inserted description of 2WA FAT, 2WA TMP, 2WA TMPxFAT in the figures.
- Is two way ANOVA an appropriate method? Which two factors are included in the analysis?
Answer: In this study, two-way ANOVA analysis is appropriate. A two-way ANOVA has two independent variables. In the present study, TMP was added to normal diet (ND) and high-fat diet (HD), respectively. Therefore, fat diet and TMP in this study are two independent variables. In addition, the analysis was advised by statistics experts.
Minor points
- Fig 3, Should the unit on the vertical axes (LIVER) be mg/g body weight?
Answer: Yes, we inserted the unit of vertical axes of LIVER in the Fig. 3.
- Table 4 legend “Experimental diets were adjusted form considering of protein contents.” Please rewrite this sentence to make it understandable.
Answer: We deleted the sentence in the Table 4. We inserted one sentence in materials & methods section (sect 4.2): “All groups received AIN-76 standard diet with 20% protein. SP and TMP groups were added by 20% SP and 20% TMP in the normal and high-fat diet, respectively for 12 weeks. Composition of the diets is presented in Table 4.

Reviewer 2 Report
This manuscript evaluated Tenebrio molitor larva fermentation extract as a potential substitute protein for soy protein using high-fat diet–induced obese mice. Non-alcoholic fatty liver and insulin resistance are the most representative metabolic disorder. Here, the authors proposed a good solution in this field. I think authors’ hypothesis and their result are encouraging. Therefore, I will recommend this manuscript is worth to publish in your journal. However, there are a few minor comments. 1. In this article, the authors refer to HF for ‘high-fat diet’ and ND for ‘normal diet.’ But in fact, ‘high-fat’ is in the whole manuscript. I recommend ‘high-fat’ to ‘HD.’ 2. The authors should explain correct about hepatoprotective effects of extracts of insects in previous studies. Forsythia viridissima is a species of flowering plant in the genus Forsythia, native to southern China and South Korea, and introduced to Japan and the United States. 3. In the Animal experiments, supplied levels or amounts of ‘TMP’ and ‘SP’ were not stated in the sentences. I recommended their supplementation amount should be in manuscripts. 4.. Figure 1 needs to include more details. ‘WAT’→’WAT weight’ 5. Legend of figure 3: please insert ‘Histological analysis magnification 200x.’ 6. Many spellings need to be corrected in the manuscript overall. (e.g Gryllus bimaculatu→ Gryllus bimaculatus , higer → higher, data now shown → data not shown etc.) 7. Authors seem to be well- described to OS (oxidative stress) and ER stress with their results. Probably, the article ‘ER Stress Is Implicated in Mitochondrial Dysfunction-Induced Apoptosis of Pancreatic Beta Cells’ (published 2010) can be helpful to authors’ explanation even though the paper was about pancreatic b-cell. And I recommend an in vitro test using TMP for mitochondria dysfunction as your further study, if authors are interested in HYOU1.Author Response
Revisions and statement made according to
Reviewer2 response and comments
Thank you very much for your kind review and valuable suggestions. We have modified the manuscript accordingly, and detailed corrections are listed below point by point:
- In this article, the authors refer to HF for ‘high-fat diet’ and ND for ‘normal diet.’ But in fact, ‘high-fat’ is in the whole manuscript. I recommend ‘high-fat’ to ‘HD.’.
Answer: We revised ‘high-fat’ to ‘HD’ and highlighted in entire manuscript.
- The authors should explain correct about hepatoprotective effects of extracts of insects in previous studies. Forsythia viridissima is a species of flowering plant in the genus Forsythia, native to southern China and South Korea, and introduced to Japan and the United States.
Answer: The ‘Forsythia viridissima’ was deleted from in Page 1, line 11.
- In the Animal experiments, supplied levels or amounts of ‘TMP’ and ‘SP’ were not stated in the sentences. I recommended their supplementation amount should be in manuscripts.
Answer: Thank you for your helpful comments. We inserted one sentence in materials & methods (section 4.2): “All groups received AIN-76 standard diet with 20% protein. SP and TMP groups were added by 20% SP and 20% TMP in the normal and high-fat diet, respectively for 12 weeks. Composition of the diets is presented in Table 4.”
- Figure 1 needs to include more details. ‘WAT’→’WAT weight’,
Answer: We revised the details to ‘WAT → WAT weight’ in Figure 1.
- Legend of figure 3: please insert ‘Histological analysis magnification 200x.’
Answer: One sentence was inserted with ‘Histological analysis magnification 200x.’ in Figure 3.
- Many spellings need to be corrected in the manuscript overall. (e.g Gryllus bimaculatu→ Gryllus bimaculatus, higer → higher, data now shown → data not shown etc.).
Answer: As you suggested, we have carefully revised our errors through the full manuscript.
- Page 1, line 12: Gryllus bimaculatu → Gryllus bimaculatus
â‘¡ Page 4, line 3: higer → higher
â‘¢ Page 5, line 4: hutUH → hutU
â‘£ Page 5, line 4: hsd71 → hsc71
⑤ Page 6, line 3: artherogenic → atherogenic
â‘¥ Page 10, line 6: data now shown → data not shown
⑦ Page 10, line 16: PAPRγ → PPARγ
â‘§ Page 10, line 25: pervious → previous
⑨ Page 14, line 2 from bottom: ccontaining → containing
- Authors seem to be well- described to OS (oxidative stress) and ER stress with their results. Probably, the article ‘ER Stress Is Implicated in Mitochondrial Dysfunction-Induced Apoptosis of Pancreatic Beta Cells’ (published 2010) can be helpful to authors’ explanation even though the paper was about pancreatic b-cell. And I recommend an in vitro test using TMP for mitochondria dysfunction as your further study, if authors are interested in HYOU1.
Answer: Thanks for the good comment. The experiment conducted by Lee et al. (2010), which you proposed as a reference, is very interesting. In the future study, we will try to be able to explain with in vitro test using TMP for mitochondrial dysfunction.

Reviewer 3 Report
The manuscript by Ham and co-workers intended to investigate whether defatted mealworm fermentation extract (TMP) can be utilized as a protein substitute and used as replacement for soy protein (SP). To accomplish their goals, authors evaluated the effects of TMP substitution in a normal diet or a high-fat diet of mice for 12 weeks. Overall, the rational of the study looks interesting and results can be of general interest. However, there is a number of major and minor points that would need to be addressed in order to improve the quality of this paper before it can be accepted for publication:
- Although authors performed a detailed description of the methods, there are some points missing. Please provide the catalog numbers for the high fat diet, all the kits, antibodies and all other reagents. This is essential for the reproducibility of the data.
- Please include further information about the concentration of TMP used in the study and how it was introduced in the diet of the animals.
- Looking at the experimental groups, it seems that is missing a Control group of mice under a standard diet, i.e., without any manipulation. This would be quite important to fully understand the outcomes of the study. Please provide a coherent justification on this.
- Figures should include the number of animals used in the experiments.
- The use of ab letters to indicate significant differences among the experimental groups is quite confusing. Please remove this and just use the “*” symbols to indicate differences whenever existing.
- Please consider to complement the gene expression data with protein expression data; authors are encouraged to perform western blot to assess if TMP supplementation alters the levels of key proteins.
Author Response
Revisions and statement made according to
Reviewer 3 response and comments
Thank you very much for your kind review and valuable suggestions. We have modified the manuscript accordingly, and detailed corrections are listed below point by point:
- Although authors performed a detailed description of the methods, there are some points missing. Please provide the catalog numbers for the high fat diet, all the kits, antibodies and all other reagents. This is essential for the reproducibility of the data.
Answer: With plenty considering your comments, we revised to sentences in materials and methods section as follows:
â‘ We was unable to provide the catalog number of the high-fat diet. We were manufactured directly experimental diets based on the AIN-76 rodent diet.
â‘¡ 4.4. Serum parameters (line 1-4): “The serum insulin (M1104, Morinaga Institute of Biological Science, Inc., Yokohama, Japan), tumor necrosis factor-α (TNF-α; BMS607HS, Invitrogen, Carlsbad, CA, USA), interleukin-6 (IL-6; BMS603HS, Invitrogen), adiponectin (DY1119, R&D Systems, Minneapolis, MN), and leptin levels (DY498, R&D Systems) were determined using an enzyme-linked immunosorbent assays (ELISA) kit.”
â‘¢ 4.7. RNA isolation and quantitative real-time PCR analysis (line 1-8): “The total RNA was isolated from the liver tissue using a TRIzol reagent (15596018, Invitrogen) according to the manufacturer's instructions. The RNA concentration and purity were evaluated using a Nanodrop 2000 spectrophotometer (Thermo Fisher Scientific, Waltham, MA, USA). The complementary DNA (cDNA) was synthesized from the total RNA (1 μg) using a ReverTra Ace qPCR RT master mix (FSQ-201, Toyobo, Osaka, Japan). Real-time PCR was performed using an SYBR green PCR kit (204143, Qiagen, Hilden, Germany) with a specific primer (Table S1) in a CFX96 TouchTM re-al-time PCR detection system (Bio-Rad Laboratories, Inc., Hercules, CA, USA).”
â‘£ 4.8. Protein sample preparation (line 1-6): “The liver tissues were homogenized directly using motor driven homogenizer (PowerGen125, Fisher Scientific, Pittsburgh, PA) in sample lysis solution containing 7 M urea (Sigma-Aldrich, St. Louis, MO, USA), 2 M thiourea (Sigma-Aldrich) containing 4% (w/v) 3-[(3-cholamidopropy)dimethyammonio]-1-propanesulfonate (CHAPS, Sigma-Aldrich), 1% (w/v) dithiothreitol (DTT, Sigma-Aldrich), 2% (v/v) pharmalyte (Amersham Biosciences, Little Chalfont , UK), and 1 mM benzamidine (Sigma-Aldrich).”
⑤ 4.9. Two-dimensional polyacrylamide gel electrophoresis (2-DE) & Peptide Mass Fingerprinting (PMF) (line 1-7): 2-DE was conducted as previous described [51]. For 2-DE analysis, immobilized pH gradient (IPG) strips (Genomine Inc., Pohang, Korea) were rehydrated in swelling buffer containing 7 M urea (Sigma-Aldrich), 2 M thiourea (Sigma-Aldrich), 1% (w/v) DTT (Sigma-Aldrich), and 4% (w/v) CHAPS (Sigma-Aldrich). Isoelectric focusing (IEF) was performed at 20°C. The gels were stained coomassie G250, and then destained using deionized water and images were acquired with an image scanner (BioRad, Hercules, USA) as described by Anderson et al.[31].
- Please include further information about the concentration of TMP used in the study and how it was introduced in the diet of the animals. Looking at the experimental groups, it seems that is missing a Control group of mice under a standard diet, i.e., without any manipulation. This would be quite important to fully understand the outcomes of the study. Please provide a coherent justification on this.
Answer: Thank you for your helpful comments. In this study, soy protein was used as a control protein. The purpose of our research is to evaluate the possibility of TMP as an alternative dietary protein. Soy protein is already known to be effective against metabolic diseases such as obesity, diabetes, and dyslipidemia; however, it has a low essential amino acids contents. We previously confirmed that phenotype (i.e. body weight, fat size, and serum levels of glucose, insulin and leptin) of obese mice improved as the amount of TMP substituted for soy protein increased. Therefore, we tried to compare the effects of TMP and soy protein in lean and obese mice. Therefore, we first compared the effects of TMP to so protein in lean and obese mice.
We inserted one sentence in materials & methods section (section 4.2): “All groups received AIN-76 standard diet with 20% protein. SP and TMP groups were added by 20% SP and 20% TMP in the normal and high-fat diet, respectively for 12 weeks. Composition of the diets is presented in Table 4.
- Figures should include the number of animals used in the experiments.
Answer: As your comments, we inserted the number of animals. “The values are expressed as the means ± S.E. (n=10 per group).”
- The use of ab letters to indicate significant differences among the experimental groups is quite confusing. Please remove this and just use the “*” symbols to indicate differences whenever existing.
Answer: Thank you for pointing this out. We indicated significant differences among the experimental groups with the ‘†’ symbol instead of the ‘ab’ letter.
- Please consider to complement the gene expression data with protein expression data; authors are encouraged to perform western blot to assess if TMP supplementation alters the levels of key proteins.
Answer: In the future, as in your opinion, I would like to analyze the gene and protein expression of key proteins. Thanks again for your kind advise.

Round 2
Reviewer 3 Report
The authors have addressed my comments.
Author Response
Thank you very much for your kind review and valuable suggestions.